# Decline of Lung Function in Knee and Spine Osteoarthritis in the Korean Population: Cross-Sectional Analysis of Data from the Korea National Health and Nutrition Examination Survey

**DOI:** 10.3390/healthcare10040736

**Published:** 2022-04-15

**Authors:** Seong-Kyu Kim, Sang Gyu Kwak, Jung-Yoon Choe

**Affiliations:** 1Division of Rheumatology, Department of Internal Medicine, Catholic University of Daegu School of Medicine, Daegu 42472, Korea; jychoe@cu.ac.kr; 2Department of Medical Statistics, Catholic University of Daegu School of Medicine, Daegu 42472, Korea; sgkwak@cu.ac.kr

**Keywords:** osteoarthritis, spine, knee, COPD, spirometry

## Abstract

Background: Evidence on the close association between osteoarthritis (OA) and lung diseases is supported by the shared pathogenesis of the two diseases. We assessed the association between knee and spine OA and chronic obstructive pulmonary disease (COPD) in the Korean population. Methods: Using data from the Korea National Health and Nutrition Examination Survey (KNHANES) 2012, a total of 2006 subjects who underwent both plain radiography for assessment of knee and lumbar spine and spirometry analysis for lung function were analyzed. Radiographic severity grade for OA was assessed using the Kellgren–Lawrence (K-L) grading scale. COPD was defined as a ratio of forced expiratory volume in one second (FEV_1_) to forced vital capacity (FVC) less than 0.7. Results: Subjects with spine OA had higher prevalence of COPD than controls (*p* < 0.001), but not knee OA (*p* = 0.990). FVC (L), FEV_1_ (L), and FVC/FEV1 (%) were significantly decreased in spine OA compared to in controls (*p* = 0.003, *p* < 0.001, and *p* < 0.001, respectively). FVC (L), FVC (%), FEV_1_ (L), and FEV_1_ (%) were significantly different between knee OA and controls. Univariate regression analysis showed that spine OA was significantly associated with COPD (OR 1.581, 95% CI 1.204–2.076, *p* = 0.001), but not knee OA. Multivariate analysis revealed that spine OA lost statistical significance for COPD. Conclusion: This study found that subjects with knee OA and spine OA had a decline of lung function compared to subjects without OA, although OA was not associated with COPD.

## 1. Introduction

Osteoarthritis (OA) is one of most common degenerative joint diseases and leads to increased joint pain and functional disability, with substantial health burden [1,2]. Due to the aging and increased population with obesity, there has been global increase in the prevalence of OA. It is estimated that 40% of men and 47% of women experience symptomatic OA during their lifetime [3]. Clinically, the knee is most commonly affected, followed by hand and hip involvement [3]. In addition, the prevalence of spine OA, presenting with degenerative changes in spinal structures, including vertebral bodies, intervertebral discs, and zygapophyseal joints, is estimated to be 20% to 85% in the elderly population [4,5,6,7,8].

Recently, interest in the comorbidities of OA patients is growing due to the potent effect of comorbidities on the clinical course and prognosis of OA. In patients with OA, 67% have one or more comorbid conditions, including hypertension, dyslipidemia, diabetes mellitus, depression, and chronic obstructive pulmonary disease (COPD) [9,10,11]. Several studies have demonstrated that lung disease, including COPD and bronchial asthma, is one of the most prevalent comorbidities in OA, and its prevalence widely ranged from 6.1% to 18.6%, due to differences in the characteristics of study populations and disease definitions [9,10,11].

COPD is characterized by respiratory symptoms of dyspnea, cough, and sputum and irreversible airflow limitations related to chronic inflammation, and is now considered a leading cause of disease-related morbidity and mortality, which is directly attributed to cigarette smoking and other risk factors, including occupational or environmental pollutants [12,13]. COPD is a complex and heterogenous disorder and, thus, could present diverse extrapulmonary comorbidities, including musculoskeletal dysfunction, cardiovascular disease, diabetes mellitus, osteoporosis, and depression [14]. In addition, OA is an important comorbid condition in patients with COPD, with an estimated prevalence ranging from 12% to 74% [15,16].

Although the primary pathogenic mechanisms of OA and COPD are different, it can be inferred that the two disease entities have clinical relevance to one another in that chronic inflammation might be a common pathogenic role in the development of the two diseases [15]. However, data on the linkage between OA of the knee and spine and COPD are limited. The purpose of this study was to determine whether radiographic knee and spine OA is associated with COPD using Korean population-based health and nutrition survey data.

## 2. Subjects and Methods

### 2.1. Study Population

The Korea National Health and Nutrition Examination Survey (KNHANES) is a national cross-sectional survey for population-based health and nutrition for non-institutionalized Korean subjects, which is annually conducted by the Korea Centers for Disease Control and Prevention (KCDC). A stratified multi-stage cluster probability sampling design was applied to reflect information on the entire Korean population.

A total of 8518 participants registered in the KNHANES 2012 were initially enrolled in this analysis (Figure 1). Among them, 3252 participants over the age of 50 underwent plain radiography examination for the knee and lumbar spine, and 3240 participants over 40 years of age underwent lung function tests using spirometry. A total of 2006 subjects who underwent both plain radiography and lung function tests were used in the study analysis.

### 2.2. Clinical Information

General characteristics of the study population were collected including age (years), sex, body mass index (BMI, kg/m^2^), and smoking status (non-smokers, ex-smokers, and current smokers), alcohol consumption status (non-alcoholics and alcoholics). The presence of comorbidities was defined through a standardized questionnaire, such as “Have you ever been diagnosed with hypertension by a physician?”. Comorbid conditions included hypertension, dyslipidemia, cerebral infarction, myocardial infarction/angina, pulmonary tuberculosis, bronchial asthma, and diabetes mellitus. The answers to the questionnaire consisted of number 1 for “Yes”, number 2 for “No”, number 8 for “Children (Not applicable)”, and number 9 for “Don’t know”. If a participant answered number 1 for “Yes” in the questionnaire for an individual comorbid condition, it was determined that they had the disease. A well-trained professional interviewer conducted in-depth interviews with each participant.

### 2.3. Radiographic Assessment

After obtaining anteroposterior and lateral plain radiography of the knee and lumbar spine, radiographic severity for knee and lumbar spine based on Kellgren-Lawrence (K-L) criteria was evaluated for participants over 50 years of age [17]. Radiologic gradings of the lumbar spine were classified as grade 0 (normal), grade 1 (suspicious; osteophyte), or grade 2 (abnormal; intervertebral disc space narrowing, bone sclerosis, or large osteophyte) [18]. Spine OA was defined as radiographic grading of K-L grade 2. The radiologic grade of the knee was assessed according to the K-L grading system, as follows: grade 0 (normal), grade 1 (doubtful), grade 2 (mild), grade 3 (moderate), or grade 4 (severe) [17]. K-L grade ≥2 of the knee was diagnosed as knee OA. In this study, participants were divided into three groups, as follows: controls (subjects without any joint involvement), knee OA group, and spine OA group.

### 2.4. Lung Function Tests

For subjects over 40 years of age, spirometry lung function tests (Model 1022 Digital Computed Spirometry^®^, Sensor Medics, Yorba Linda, CA, USA) were performed to measure forced expiratory volume in one second (FEV_1_) and forced vital capacity (FVC). If FEV_1_/FVC < 0.7, this was defined as COPD, based on the Global Initiative for Chronic Obstructive Lung Disease (GOLD) guidelines [19]. COPD staging was classified into four stages, as follows: 1 (mild, FEV_1_ ≥ 80% of the predictive value), 2 (moderate, 50 ≤ FEV_1_ ≤ 79%), 3 (severe, 30 ≤ FEV_1_ ≤ 49%), or 4 (very severe, FEV_1_ <30% or <50% with chronic symptoms).

### 2.5. Statistical Analysis

As described in our previous studies [20,21], the weighted values of the health examination survey (wt_itvex), the variance estimate layer (kstrata), and the number of enumeration districts (psu) were used, because the KNHANES data were designed as a stratified multi-stage cluster probability sampling model.

Data are described as the non-weighted number of cases (weighted %) for qualitative variables or mean (standard error, SE) for quantitative variables in the descriptive and frequency analyses. Composite sample Chi-square analyses for qualitative variables and two-sample t-tests for quantitative variables were performed. An analysis of variance (ANOVA) test was used to compare the difference in lung function results according to K-L grades for knee and spine OA. To determine the variables related to COPD, composite sample binary logistic regression analyses were applied for multivariate analysis using significant variables in univariate analysis. The results are described as odds ratio (OR) and 95% confidence interval (CI). Statistical analyses were conducted using IBM SPSS Statistics 19.0 software (IBM Corp., Armonk, NY, USA). Results are considered statistically significant if the *p* value is less than 0.05.

## 3. Results

### 3.1. Baseline Characteristics of Study Population

Baseline demographic data and information for comorbidities are presented in Table 1. The mean age of enrolled subjects was 59.2 (SE 0.2) years old. Among them, 1088 (50.3%) subjects were female. The mean BMI was 24.2 kg/m^2^ (SE 0.1). In addition, the data of smoking status, alcohol consumption, and comorbid diseases is also illustrated in Table 1. Regarding the prevalence of comorbidities, the highest frequency was in the order of hypertension (32.9%), dyslipidemia (18.3%), COPD (16.4%), and diabetes mellitus (12.9%).

### 3.2. Comparison of Clinical Variables between OA and Controls

Of the 2006 subjects included in the study, 414 subjects (18.7%) were classified as knee OA, 394 subjects (19.2%) as spine OA, and 1198 subjects (62.1%) as controls (Table 2). Subjects with knee and spine OA were older than controls (*p* < 0.001 for both). Female subjects were more prevalent in knee OA than controls (*p* < 0.001), whereas there was no difference in gender between spine OA and controls. Subjects with knee OA showed higher BMI than controls (*p* < 0.001). Considering smoking status, a significant difference between knee OA and controls was found (*p* < 0.001), whereas there was no difference in smoking status between the spine OA and control group.

Among the comorbid diseases, there was a difference between knee OA and controls in hypertension, myocardial infarction/angina, and bronchial asthma. The frequency of hypertension, cerebral infarction, myocardial infarction/angina, bronchial asthma, and diabetes mellitus in spine OA was different from controls. Subjects with spine OA had a higher frequency of COPD compared to controls (22.1% vs. 15.0%, *p* < 0.001), whereas there was similar frequency of COPD between knee OA and controls (*p* = 0.990).

### 3.3. Comparison of Clinical Variables According to Presence of COPD

Subjects with COPD showed older age and lower BMI compared to subjects without COPD (*p* < 0.001 for both) (Table 3). COPD was more common in males than females. Smoking exposure, hypertension, pulmonary tuberculosis, and bronchial asthma was more frequent in subjects with COPD than those without COPD. Spine OA was more common in subjects with COPD than in those without COPD (*p* < 0.001), whereas there was no difference in knee OA between the two groups (*p* = 0.876).

### 3.4. Comparison of Lung Function Tests between OA and Controls

Subjects with spine OA had lower FVC (L), FEV_1_ (L), and FEV_1_/FVC values compared to controls (*p* = 0.003, *p* < 0.001, and *p* < 0.001, respectively) (Table 4). Knee OA subjects showed lower FVC (L), FVC (%), and FEV_1_ (L) than controls (*p* < 0.001, *p* = 0.031, and *p* < 0.001, respectively). However, FEV_1_ (%) in subjects with knee OA was higher than in those without OA (*p* = 0.045).

There was no difference in the frequency of COPD in subjects with knee OA compared with controls (*p* = 0.523). There was a significant difference among the four stages according to COPD stage by GOLD guidelines between controls and spine OA (*p* < 0.001).

In the comparison of lung function tests according to radiographic grade, FVC (L) and FEV_1_ (L) in knee OA gradually decreased with increased K-L grade (*p* < 0.001 for both) (Figure 2), while FEV_1_/FVC (%) tended to show increases. In the lung function in spine OA, FVC (L), FEV_1_ (L), and FEV_1_/FVC (%) showed a decreasing trend with increasing K-L grade.

### 3.5. Determination for Variables Related to COPD

To identify variables related to COPD, a binary logistic regression analysis using demographic and clinical variables was performed (Table 5). Univariate analysis showed a significant association of COPD with older age, male sex, lower BMI, smoking exposure, and subjects with hypertension, pulmonary tuberculosis, and bronchial asthma. In evaluating whether OA according to the affected joints was linked to COPD, lumbar spine OA was related to COPD (OR 1.581, 95% CI 1.204–2.076, *p* = 0.001), but not knee OA (OR 1.062, 95% CI 0.765–1.475, *p* = 0.716). In the multivariate logistic regression analysis, which considers variables significant in univariate analysis as confounding factors, older age, male sex, lower BMI, smoking exposure history, pulmonary tuberculosis, and bronchial asthma were significantly associated with COPD. In contrast, spine OA lost statistical significance after adjustment with confounding factors (OR 1.216, 95% CI 0.869–1.701, *p* = 0.253).

## 4. Discussion

OA can generally be accompanied by diverse comorbid diseases, which somewhat affect the prognosis and clinical course of the disease. In particular, lung disease is one of common coexisting conditions that is complicated in patients with OA. It is well established that OA and COPD are mutual comorbidities [9,10,11,14,15,16]. Most studies have been limited to OA affecting peripheral joints, such as hand, knee, and/or hip joints. Although spine OA is a relatively common degenerative arthritis, especially in the elderly population over 50 years old, few studies have focused on the effects of spine OA on lung disease-related function disability and comorbidities. In this study, we investigated whether there is relationship between COPD, measured by spirometry and OA of the spine and knee, and confirmed by radiologic evaluation using data obtained from the KNHANES. The main finding of this study revealed that neither knee nor spine OA is associated with COPD, although subjects with knee and spine OA showed more highly impaired lung function than those without OA.

Although the similarities in the pathogenic mechanisms between OA and COPD have not been clearly elucidated, several possible hypotheses for pathogenic linkage between the two diseases have been proposed. First, sustained low-grade inflammation has been suspected in the pathogenesis of both diseases. The systemic inflammatory response mediated by pro-inflammatory cytokines such as interleukin-1 (IL-1), IL-6, IL-18, and transforming growth factor-β or collagen destructive enzymes, such as matrix metalloproteinase-9 (MMP-9), has been found to be responsible for aggravation of OA and COPD [22,23]. Pro-inflammatory cytokines were highly expressed in COPD patients and were significantly associated with the severity of lung function [24]. Similarly, high expression of MMP-2 and MMP-9, contributing to cartilage destruction, was also detected in the serum, synovial fluid, and synovial tissue of OA patients [25,26], suggesting that inflammatory or immune response in both diseases appears to lead to damage of the articular structures and lung tissues. Second, OA and COPD share similar risk factors related to the exacerbation or onset of the disease, particularly aging. The prevalence of OA has a linear increase at 50 years of age and greater. In addition, most adults over 65 years of age show OA-related radiographic changes in at least one joint of the knee and/or spine [27,28]. Lung function also gradually decreases with increasing age, although elderly adults might not be aware of the decline in respiratory function [29]. The exact mechanism and whether aging is a common risk factor for development and progression of the two diseases has not been determined. Reactive oxygen species (ROS) produced through diverse cellular pathways might contribute to the acceleration of aging itself, as well as the development of age-related diseases [30]. ROS are toxic molecules and are a main mediator of oxidative stress that causes direct damage to target tissues and cells. Aging-related oxidative stress is considered one of the main determinants in the pathogenesis of OA and COPD [23,31]. Based on this evidence, the development or progress in the two diseases might be attributable to aging. We also confirmed that older age was equally related with COPD, as well as OA of the knee and spine, in this study population (Table 5 and Appendix A).

The effect of gender on COPD development and progression is controversial. Men are known to have a higher risk of COPD, due to a higher frequency of cigarette smoking or multiple occupational exposures. Our study also found that male sex was significantly associated with COPD. In contrast, women have recently been found to be more vulnerable to COPD than men when exposed to the same dose of inhaled irritants, such as smoking [32]. Mice exposed to chronic cigarette smoke induced emphysematous-like changes in lung tissue, which led to more rapid changes in females than in males [33], suggesting that lung tissue in females is more susceptible to oxidant stress caused by cigarette smoke. Considering the gender effect on the vulnerability to OA, deficiency of estrogen contributed to aggravation of damage of cultured articular chondrocytes by oxidative stress [34], which is evidence that incidence and prevalence of menopause-related OA increase in females. In many epidemiological studies, the higher prevalence of OA in women than in men suggests evidence of a sex effect [2]. However, this study found that male subjects were significantly associated with COPD. In contrast, female gender was closely linked to knee OA compared to male, whereas there was no gender difference between spine OA and controls (Appendix A). We found that the gender effect on COPD and OA of the knee and spine had a significant disparity. Therefore, gender cannot be considered a shared risk factor for COPD or OA. More studies on the effect of sex on susceptibility to these diseases are needed.

The association between BMI and lung function or COPD is uncertain. In an analysis of 3631 COPD patients from a prospective cohort study on the genetic epidemiology of COPD, obesity was found to be associated with significantly worse COPD-related outcomes and poor quality of life [35]. In contrast, a recent meta-analysis on the obesity paradox in COPD showed that low BMI was linked with acceleration of diminished lung function, whereas a high BMI showed beneficial effects on lung function [36]. Consistent with the findings of that meta-analysis, our study also showed that lower BMI is closely associated with COPD. However, considering the debate on the relationship between BMI or obesity and COPD, additional studies are necessary.

Although this study found that neither knee OA nor spine OA were associated with COPD, there was significant differences in lung function FVC and FEV_1_ test results between subjects with knee and spine OA and controls. Additional explanations are needed for the fact that OA was not associated with COPD, despite declines in lung function in OA. Especially, this study showed that only spine OA was significantly associated with COPD in univariate analysis. First, we should consider the possibility that comorbidities in patients with spine OA may have affected the measures of lung function. In this study, the statistical significance between spine OA and COPD disappeared after adjustment with other variables, including comorbid conditions. Second, the pathologic mechanisms such as skeletal muscle wasting and weakness should be considered. It is well established that inflammation, malnutrition, physical inactivity, and metabolic dysfunction in chronic inflammatory diseases such as COPD and OA induce worsening of skeletal muscle mass and strength [37,38]. The prevalence of sarcopenia is estimated to be 4.4% to 27.5% for COPD and about 29.3% for OA [37,38]. Jeon et al. demonstrated that low skeletal muscle mass was related to knee OA in a cross-sectional study that included 3813 subjects, whereas it was negatively linked with spine OA [39]. This suggests that even if low skeletal muscle mass results in decreased lung function, the decline in lung function causes a reduction in spine OA. Furthermore, it seems that skeletal muscle wasting shows differential clinical effects, by different mechanisms, for the knee and lumbar spine. Further studies are needed to determine whether sarcopenia in knee and lumbar spine OA could affect COPD.

In addition, we found that subjects with both knee and spine OA also showed decreased lung function compared to controls (Appendix A). Moreover, lung function in patients with both knee and spine OA was more impaired than those with either knee or spine OA. This suggests that the decrease in lung function, in part, appears to be additively severe with the number of joints involved. Despite these differences in lung function, this study identified that patients with both knee and spine OA were not associated with the presence of COPD. Interestingly, radiographic progression in knee OA was found to be related with impaired lung function. However, Lee et al. demonstrated that radiographic severity of knee OA was associated with asthma, but not COPD [40]. Further studies are needed to determine whether the radiographic progression and the extent of involved joints of OA affect the severity of lung function.

There are some limitations to this study. First, there might be differences in the prevalence of radiologic OA and symptomatic OA [6,27]. Variables for symptomatic OA components, including knee and lower back pain, were not considered in this study, because questionnaires about joint symptoms usually exist in the form of self-reports, so the consistency of answers cannot be guaranteed. Second, there is no clear radiologic severity grading system for lumbar spine OA. Muraki et al. evaluated lumbar spondylosis from grade 0 to grade 4 using the K-L grading [41]. They defined lumbar spondylosis as higher than K-L grade 2 (definite osteophytosis with some sclerosis of the anterior part of the vertebral plate), which is considered similar to the K-L grade 2 used in this study. It is necessary to validate the radiologic grading system for spinal joints in future studies.

This study is the first investigation about the relationship between spine OA and COPD using data from the Korean population-based national survey. We found that COPD was markedly associated with spine OA in the univariate analysis, but the statistical significance disappeared after adjustment with confounding variables. However, this study confirmed a more significant decrease in lung function in patients with knee and lumbar spine OA than in those without OA. Additional studies focusing on the pathogenic mechanism of the close association of OA with COPD, and the causal relationship between the two disease entities, are needed.

## Figures and Tables

**Figure 1 healthcare-10-00736-f001:**
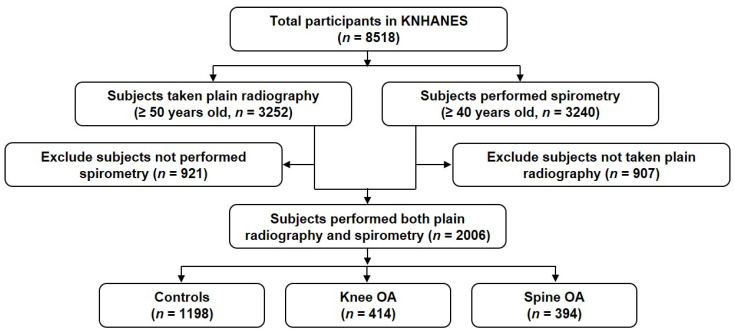
Study population flow chart. Abbreviation: KNHANES, Korea National Health and Nutrition Examination Survey; OA, osteoarthritis.

**Figure 2 healthcare-10-00736-f002:**
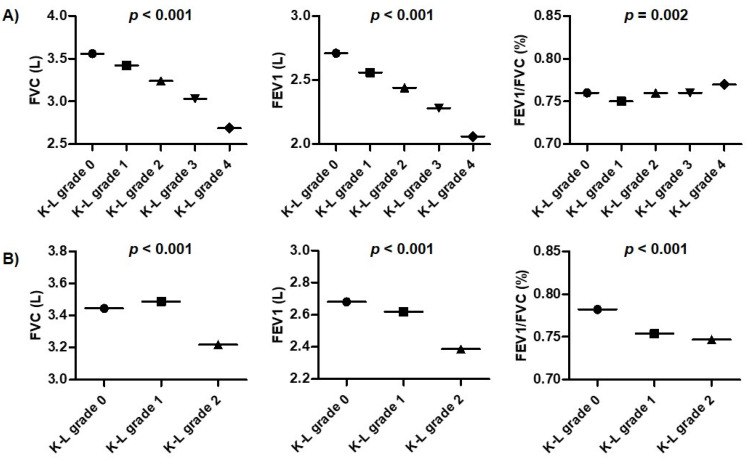
Comparison of lung function tests according to K-L grades for knee and lumbar spine. (**A**) Knee OA, (**B**) Spine OA. Data was described as mean values. *p* values were obtained by one-way ANOVA. Abbreviation: FVC, forced vital capacity; FEV_1_, forced expiratory volume in one second; OA, osteoarthritis; K-L grade, Kellgren–Lawrence grade.

**Table 1 healthcare-10-00736-t001:** Baseline characteristics of study population.

Parameters	Values
Age (years)	59.2 (0.2)
Sex (female, *n*, %)	1088 (50.3)
Body mass index (kg/m^2^)	24.2 (0.1)
Smoking (*n*, %) *	
Non-smokers	1145 (53.5)
Ex-smokers	494 (24.8)
Current smokers	348 (21.7)
Alcohol consumption (*n*, %) *	
Non-alcoholic	360 (15.5)
Alcoholic	1628 (84.5)
Comorbidities (*n*, %) *	
Hypertension	707 (32.9)
Dyslipidemia	427 (18.3)
Cerebral infarction	39 (1.9)
Myocardial infarction/angina	74 (3.5)
Pulmonary tuberculosis	163 (8.0)
Bronchial asthma	107 (5.6)
Diabetes mellitus	261 (12.9)
COPD	337 (16.4)

Data are described as non-weighted number of cases (weighted %) for qualitative variables or mean (standard error, SE) for quantitative variables. Abbreviation: FVC, forced vital capacity; FEV_1_, forced expiratory volume in one second; COPD, chronic obstructive pulmonary disease. * Missing data were excluded from the analyses: for smoking, *n* = 19; for alcohol consumption, *n* = 18; comorbidities (hypertension, dyslipidemia, cerebral infarction, myocardial infarction/angina, pulmonary tuberculosis, bronchial asthma, and diabetes mellitus), *n* = 16.

**Table 2 healthcare-10-00736-t002:** Comparison of characteristics between knee or spine OA and controls.

Parameters	Controls	Knee OA	*p* Value *	Spine OA	*p* Value *
Prevalence (*n*, %)	1198 (62.1)	414 (18.7)		394 (19.2)	
Age (years)	57.3 (0.3)	62.3 (0.5)	<0.001	62.5 (0.6)	<0.001
Sex (female, *n*, %)	604 (46.0)	285 (66.3)	<0.001	199 (49.0)	0.789
Body mass index (kg/m^2^)	23.9 (0.1)	25.3 (0.2)	<0.001	24.2 (0.2)	0.730
Smoking (*n*, %) ^†^			<0.001		0.871
Non-smokers	636 (49.1)	296 (68.6)		213 (52.8)	
Ex-smokers	317 (27.1)	72 (15.9)		105 (26.2)	
Current smokers	232 (23.8)	43 (15.5)		73 (20.9)	
Alcohol consumption (*n*, %) ^†^			<0.001		0.002
Non-alcoholic	178 (12.7)	112 (24.2)		70 (15.9)	
Alcoholic	1006 (87.3)	300 (75.8)		322 (84.1)	
Comorbidities (*n*, %) ^†^					
Hypertension	346 (27.0)	195 (44.1)	<0.001	166 (40.6)	<0.001
Dyslipidemia	254 (17.2)	95 (22.0)	0.497	78 (18.1)	0.076
Cerebral infarction	29 (2.3)	4 (0.8)	0.069	6 (1.7)	0.010
Myocardial infarction/angina	31 (2.6)	20 (3.6)	0.026	23 (6.1)	<0.001
Pulmonary tuberculosis	104 (8.4)	39 (9.3)	0.676	20 (5.4)	0.218
Bronchial asthma	50 (4.5)	32 (7.9)	0.005	25 (6.7)	0.012
Diabetes mellitus	133 (11.1)	59 (14.2)	0.097	69 (17.4)	<0.001
COPD	182 (15.0)	63 (15.0)	0.990	92 (22.1)	<0.001

Data are described as non-weighted number of cases (weighted %) for qualitative variables or mean (standard error, SE) for quantitative variables Abbreviation: FVC, forced vital capacity; FEV_1_, forced expiratory volume in one second; COPD, chronic obstructive pulmonary disease * *p* values were compared to controls and obtained by two sample *t*-test for quantitative variables or chi-square test for qualitative variables. ^†^ Missing data were excluded from the analyses: for smoking, *n* = 19; for alcohol consumption, *n* = 18; comorbidities (hypertension, dyslipidemia, cerebral infarction, myocardial infarction/angina, pulmonary tuberculosis, bronchial asthma, and diabetes mellitus), *n* = 16.

**Table 3 healthcare-10-00736-t003:** Comparison of characteristics between subjects with and without COPD.

Parameters	Subjects without COPD	Subjects with COPD	*p* Value *
Prevalence (*n*, %)	1669 (83.6)	337 (16.4)	
Age (years)	58.4 (0.2)	63.5 (0.6)	<0.001
Sex (female, *n*, %)	1007 (55.7)	81 (22.9)	<0.001
Body mass index (kg/m^2^)	24.4 (0.1)	23.3 (0.2)	<0.001
Smoking (*n*, %) ^†^			<0.001
Non-smokers	1056 (59.4)	89 (23.6)	
Ex-smokers	345 (21.8)	149 (40.1)	
Current smokers	251 (18.8)	97 (36.3)	
Alcohol intake (*n*, %) ^†^			0.302
Non-alcoholic	316 (16.2)	44 (11.9)	
Alcoholic	1336 (83.8)	292 (88.1)	
Comorbidities (*n*, %) ^†^			
Hypertension	580 (32.9)	127 (32.8)	<0.001
Dyslipidemia	370 (19.1)	57 (14.0)	0.427
Cerebral infarction	31 (1.8)	8 (2.4)	0.068
Myocardial infarction/angina	59 (3.4)	15 (3.8)	0.249
Pulmonary tuberculosis	105 (6.2)	58 (17.2)	<0.001
Bronchial asthma	68 (4.2)	39 (12.3)	<0.001
Diabetes mellitus	215 (12.9)	46 (12.7)	0.053
Knee OA	351 (19.0)	63 (17.2)	0.876
Spine OA	302 (17.9)	92 (25.9)	<0.001

Data are described as non-weighted number of case (weighted %) for qualitative variables or mean (standard error, SE) for quantitative variables Abbreviation: COPD, chronic obstructive pulmonary disease; OA, osteoarthritis * *p* values were obtained by two sample *t*-test for quantitative variables or chi-square test for qualitative variables. ^†^ Missing data were excluded from the analyses: for smoking, *n* = 19; for alcohol consumption, *n* = 18; comorbidities (hypertension, dyslipidemia, cerebral infarction, myocardial infarction/angina, pulmonary tuberculosis, bronchial asthma, and diabetes mellitus), *n* = 16.

**Table 4 healthcare-10-00736-t004:** Comparison of lung functions between subjects with OA and controls.

Parameters	Total	Controls	Knee OA	*p* Value *	Spine OA	*p* Value *
Spirometry						
FVC (L)	3.46 (0.02)	3.56 (0.03)	3.18 (0.05)	<0.001	3.41 (0.06)	0.003
FVC (%)	92.0 (0.4)	92.2 (0.48)	91.1 (0.6)	0.031	92.3 (0.7)	0.665
FEV_1_ (L)	2.62 (0.02)	2.71 (0.02)	2.42 (0.04)	<0.001	2.52 (0.05)	<0.001
FEV_1_ (%)	90.8 (0.4)	90.3 (0.5)	91.9 (0.8)	0.045	91.0 (1.0)	0.619
FEV_1_/FVC (%)	0.76 (0.00)	0.76 (0.00)	0.76 (0.00)	0.640	0.74 (0.01)	<0.001
COPD stage				0.523		<0.001
Stage 0	1669 (83.6)	1016 (85.0)	351 (85.0)		302 (77.9)	
Stage 1	146 (7.2)	78 (6.7)	28 (5.9)		40 (9.9)	
Stage 2	176 (8.5)	99 (7.9)	33 (8.7)		44 (10.3)	
Stage 3	14 (0.7)	5 (0.4)	1 (0.2)		8 (1.9)	
Stage 4	1 (0.0)	0 (0.0)	1 (0.2)		0 (0.0)	

Abbreviation: FVC, forced vital capac1ity; FEV_1_, forced expiratory volume in one second; COPD, chronic obstructive pulmonary disease; OA, osteoarthritis. * *p* values were compared to controls and obtained by two sample t-test or chi-square test.

**Table 5 healthcare-10-00736-t005:** Multivariate-adjusted ORs (95% CIs) for variables related to the presence of COPD.

Parameters	Univariate Analysis			Multivariate Analysis		
	OR	95% CI	*p* Value *	OR	95% CI	*p* Value *
Age	1.084	1.073–1.096	<0.001	1.086	1.065–1.107	<0.001
Sex (ref: female)	3.393	2.600–4.428	<0.001	2.342	1.423–3.853	0.001
Body mass index	0.903	0.864–0.945	<0.001	0.884	0.830–0.943	<0.001
Smoking (ref: non-smoker) ^†^						
Ex-smoker	3.883	2.894–5.212	<0.001	2.234	1.358–3.677	<0.001
Current smoker	3.742	2.707–5.171	<0.001	3.426	1.986–5.908	0.038
Hypertension (ref: none) ^†^	1.656	1.299–2.110	<0.001	0.910	0.645–1.284	0.588
Pulmonary tuberculosis (ref: none) ^†^	3.714	2.577–5.353	<0.001	3.277	1.976–5.434	<0.001
Bronchial asthma (ref: none) ^†^	3.389	2.192–5.242	<0.001	3.022	1.934–4.722	<0.001
Spine OA (ref: controls)	1.581	1.204–2.076	0.001	1.216	0.869–1.701	0.253
Knee OA (ref: controls)	1.062	0.765–1.475	0.716			

Abbreviation: OR, odds ratio; CI, confidence interval; OA, osteoarthritis. * *p* values were obtained by the composite sample multivariate logistic regression analysis. ^†^ Missing data were excluded from the analyses: for smoking, *n* = 19; for alcohol consumption, *n* = 18; comorbidities (hypertension, pulmonary tuberculosis, and bronchial asthma), *n* = 16.

## Data Availability

Data are available on the official KNHANES website (https://knhanes.kdca.go.kr/knhanes/sub03/sub03_02_05.do, accessed on 1 January 2021).

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
