# Peer review of "Decline of Lung Function in Knee and Spine Osteoarthritis in the Korean Population: Cross-Sectional Analysis of Data from the Korea National Health and Nutrition Examination Survey"

_healthcare, 2022, doi:10.3390/healthcare10040736_

Round 1
Reviewer 1 Report
Data of the OA prevalence should be given worldwide of for the Korean population. References are based on US population.
A Figure to display workflow for sample selection and analysis would be showed.
In OA, a patient with KL = 1 is considered as he/she has OA. Why did you only considered patients with KL >= 2?
Statistical analysis would be better explained. Did you check normality on the data before apply t-test? Did you analyse the statistical differences between the means of the three independent group or between two independent groups? If that, Mann-Whitney U test should be used rather than ANOVA. Moreover, all analysis are done using both, OA and COPD as outcome variable. However, multivariate analysis has been just done using the latter. Is there any explanation for it?
Author Response
Dear Editor
Manuscript ID: healthcare-1655875
Type of manuscript: Article
Title: Decline of lung function in knee and spine osteoarthritis in Korean population: Cross-sectional analysis of data from the Korea National Health and Nutrition Examination Survey
Thank for the editor and reviewers of the ‘Healthcare’ for reviewing our manuscript. We have made some corrections and clarifications in the revised manuscript according to the editor’s or reviewer's comments. You can find out tracing marks for changes in revised manuscript. The changes are summarized below:
Reviewer: 1
- Data of the OA prevalence should be given worldwide of for the Korean population. References are based on US population.
Answer)
We agree with your opinion. Therefore, two additional references related with spine OA prevalence in Korean population are added in the revised manuscript as follows.
- Ko, S.; Vaccaro, A.R.; Lee, S.; Lee, J.; Chang, H. The prevalence of lumbar spine facet joint osteoarthritis and its association with low back pain in selected Korean populations.Clin. Orthop. Surg. 2014, 6, 385-391.
- Cho, H.J.; Morey, V.; Kang, J.Y.; Kim, K.W.; Kim, T.K. Prevalence and Risk Factors of Spine, Shoulder, Hand, Hip, and Knee Osteoarthritis in Community-dwelling Koreans Older Than Age 65 Years.Clin. Orthop. Relat. Res. 2015, 473, 3307-3314.
- A Figure to display workflow for sample selection and analysis would be showed.
Answer)
Thanks for your comment. We add a figure (Fig. 1) for workflow for sample selection with a figure legend, as shown below.
- In OA, a patient with KL = 1 is considered as he/she has OA. Why did you only considered patients with KL >= 2?
Answer)
Thanks for your kind comment. First of all, there are no established diagnostic criteria for radiographic osteoarthritis in the spine. The radiological criteria for the raw data from Korea National Health and Nutrition Examination Survey (KNHANES) were classified by Yoshimura et al. [reference No. 16]. Even in case of suspicious or doubtful radiographic changes (grade 1) in knee osteoarthritis, it is not classified as knee osteoarthritis similar to spine OA.
- Statistical analysis would be better explained. Did you check normality on the data before apply t-test? Did you analyze the statistical differences between the means of the three independent group or between two independent groups? If that, Mann-Whitney U test should be used rather than ANOVA.
Answer)
Thanks for your valuable comment. The KNHANES was designed using a complex, stratified, multistage probability-sampling model, and data were analyzed via the complex-sample design to represent the prevalence in the Korean national population. The values of KNHANES data are representative of Korean population through the population selected from all Koreans. Therefore, the Korea Centers for Disease Control and Prevention (KCDC) provides raw data under the assumption that all data satisfies the normal distribution. All analysis should be performed according to their analysis guidelines, so arbitrarily non-parametric analysis cannot be performed. All previous studies using KNHANES data provided from the KCDC have been analyzed under the assumption of a normal distribution.
- Moreover, all analyses are done using both, OA and COPD as outcome variable. However, multivariate analysis has been just done using the latter. Is there any explanation for it?
Answer)
Although there have been studies that the lung function is decreased in patients with OA, there are few studies on the effect of knee and/spine OA on the lung function. Therefore, the main purpose of this study was to determine whether OA affects the prevalence of COPD or decreased lung function. Therefore, we analyzed after setting COPD as a dependent variable and clinical parameters including OA as independent variables. In contrast, a supplementary file showed that COPD might influence knee and spine OA.
Reviewer 2 Report
Big number of subjects, relevand and specific data, very detailed description of the results, good statistics.
New data about the relationship between OA and COPD, a start base for future studies in this field, considering the health impact for both diseases.
Author Response
Dear Editor
Manuscript ID: healthcare-1655875
Type of manuscript: Article
Title: Decline of lung function in knee and spine osteoarthritis in Korean population: Cross-sectional analysis of data from the Korea National Health and Nutrition Examination Survey
Thank for the editor and reviewers of the ‘Healthcare’ for reviewing our manuscript. We have made some corrections and clarifications in the revised manuscript according to the editor’s or reviewer's comments. You can find out tracing marks for changes in revised manuscript. The changes are summarized below:
Reviewer: 2
Big number of subjects, relevant and specific data, very detailed description of the results, good statistics. New data about the relationship between OA and COPD, a start base for future studies in this field, considering the health impact for both diseases.
Answer)
We thank the reviewers for their thoughtful and undeserved comments.
Reviewer 3 Report
The authors investigated the association between knee and spine OA and COPD. Although OA patients demonstrated decline in lung function, no association was found related to knee OA. Spine OA was significantly associated with COPD, however, this association was lost during multivariate analysis.
Comments.
- All the typos should be corrected.
- Abstract, Line 26: The authors should indicate whether “marked” means significant decline. This should be corrected.
- Lines 73, 92, 102: The patients age is not clear. This should be clarified.
- Figure 1 is missing. This should be corrected.
Author Response
Dear Editor
Manuscript ID: healthcare-1655875
Type of manuscript: Article
Title: Decline of lung function in knee and spine osteoarthritis in Korean population: Cross-sectional analysis of data from the Korea National Health and Nutrition Examination Survey
Thank for the editor and reviewers of the ‘Healthcare’ for reviewing our manuscript. We have made some corrections and clarifications in the revised manuscript according to the editor’s or reviewer's comments. You can find out tracing marks for changes in revised manuscript. The changes are summarized below:
Reviewer 3
The authors investigated the association between knee and spine OA and COPD. Although OA patients demonstrated decline in lung function, no association was found related to knee OA. Spine OA was significantly associated with COPD. However, this association was lost during multivariate analysis.
Comments.
- All the typos should be corrected.
Answer)
Thanks for kind comment. This manuscript has been proofread by a professional English proofreading company, and the typo has been verified once again.
- Abstract, Line 26: The authors should indicate whether “marked” means significant decline. This should be corrected.
Answer)
We also agree with your comment. So “marked” is deleted in the sentence.
- Lines 73, 92, 102: The patients age is not clear. This should be clarified.
Answer)
Thanks for your valuable opinion. We add a flowchart of study population at figure 1. We provide information about age in the study population like this; “The mean age of subjects performed spirometry (n = 3,240) and radiography (n = 3,252) was 57.2 (SE 0.2) and 64.1 (SE 0.2), respectively”. However, if you understand, it would be better not to describe their age in detail as they are not the actual analyzed patients.
- Figure 1 is missing. This should be corrected.
Answer)
Thanks for your comment. We already submitted a figure 1. Presumably, it seems to have been omitted during the editorial process of the paper in the journal. After revision, we attach figure 2 below for reviewer to find out. Of course, we will be sure to attach it when re-submitting.
Reviewer 4 Report
This is an interesting study. My concerns are enlisted below:
- How do the authors discuss the patients with both knee OA and spine OA ?
- How do the authors mitigate the impact of surveillance bias ? OA is a quite subjective disease, and surveillance bias is possible.
- The impact concerning the severity (K-L grading) of knee/spine OA on the results of lung function tests should be discussed.
Author Response
Dear Editor
Manuscript ID: healthcare-1655875
Type of manuscript: Article
Title: Decline of lung function in knee and spine osteoarthritis in Korean population: Cross-sectional analysis of data from the Korea National Health and Nutrition Examination Survey
Thank for the editor and reviewers of the ‘Healthcare’ for reviewing our manuscript. We have made some corrections and clarifications in the revised manuscript according to the editor’s or reviewer's comments. You can find out tracing marks for changes in revised manuscript. The changes are summarized below:
Reviewer: 4
This is an interesting study. My concerns are enlisted below:
- How do the authors discuss the patients with both knee OA and spine OA?
Answer)
Thanks for your valuable comment. We additionally present the results of lung function analysis of patients with both spine and knee OA in Supplementary table 2. And we also discuss the patients with both knee and spine OA at the end of discussion part, as follows; “In addition, we found that subjects with both knee and spine OA also showed de-creased lung function compared to controls (Supplementary table 2). Moreover, lung function in patients with both knee and spine OA was more impaired than those with either knee or spine OA. It suggests that the decrease in lung function in part appears to be as addictively severe as the number of joints involved. Even though these differences of lung function, this study identified that patients with both knee and spine OA was not associated with the presence of COPD.”.
- How do the authors mitigate the impact of surveillance bias? OA is a quite subjective disease, and surveillance bias is possible.
Answer)
Thanks for your valuable comment. Surveillance bias cannot be fundamentally blocked. However, the characteristics of the data used in this paper may be helpful in correcting this bias. The KNHANES was designed using a complex, stratified, multistage probability-sampling model, and data were analyzed via the complex-sample design to represent the prevalence in the Korean national population. The values of KNHANES data are representative of Korean population through the population selected from all Koreans. The final weight of data or results was calculated through design weight calculation, non-response rate adjustment, post-stratification, and extreme weight processing. The design weight is reflected as the reciprocal of the extraction rate to correct the surveillance bias when estimating parameters.
- The impact concerning the severity (K-L grading) of knee/spine OA on the results of lung function tests should be discussed.
Answer)
Thanks for your valuable comment. We discuss association between OA severity and lung function, at the end of discussion part, as follows; “Interestingly, radiographic progression in knee OA was found to be related with impaired lung function. However, Lee et al. demonstrated that radiographic severity of knee OA was associated with asthma but not COPD [40]. Further studies are needed to determine whether the radiographic progression and the extent of involved joints of OA affect the severity of lung function”.
Round 2
Reviewer 1 Report
Thank you for all your corrections and clarifications.
Reviewer 4 Report
My concerns have been properly addresses.